# Identification of Differentially Expressed Genes in the Longissimus Dorsi Muscle of Luchuan and Duroc Pigs by Transcriptome Sequencing

**DOI:** 10.3390/genes14010132

**Published:** 2023-01-03

**Authors:** Pengcheng Pan, Zhaoxian Qin, Wan Xie, Baojian Chen, Zhihui Guan, Bingkun Xie

**Affiliations:** 1Guangxi Key Laboratory of Livestock Genetic Improvement, Guangxi Agricultural Vocational and Technical University, Nanning 530001, China; 2College of Veterinary Medicine, Nanjing Agricultural University, Nanjing 210095, China

**Keywords:** growth and development, Luchuan pig, longissimus dorsi muscles, myosin light chain-2 (MYL2/MLC-2), transcriptomics

## Abstract

The Duroc pig originated in the United States and is a typical lean-meat pig. The breed grows fast, and the body size is large, but the meat quality is poor. The Luchuan pig is one of eight excellent local breeds in China; it has tender meat but is small in size. To study the factors that determine growth, we selected the longissimus dorsi muscle of Luchuan and Duroc pigs for transcriptome sequencing. The results of the transcriptome showed that 3682 genes were differentially expressed (DEGs) in the longissimus dorsi muscle of Duroc and Luchuan pigs. We screened out genes related to muscle development and selected the MYL2 (Myosin light chain-2) gene to perform preliminary research. Gene Ontology (GO) enrichment of biological functions and Kyoto Encyclopedia of Genes and Genomes (KEGG) analysis showed that the gene products were mainly involved in the Akt/FoxO signaling pathway, fatty acid metabolism, arachidonic acid metabolism and glycine, serine and threonine metabolism. Such pathways contributed to skeletal muscle growth, fatty acid metabolism and intramuscular fat deposition. These results provide insight into the mechanisms underlying the formation of skeletal muscle and provide candidate genes to improve growth traits, as well as contribute to improving the growth and development traits of pigs through molecular breeding.

## 1. Introduction

China is the largest pig-breeding and pork-consuming country worldwide. The yield and quality of pork are closely related to the process of muscle development, and the content of intramuscular fat is an important index for detecting the quality of pork [1]. In recent years, research on this aspect mainly concerned the complex physiological and genetic mechanisms related to intramuscular fat (IMF) deposition, gene expression pattern and its interaction in the development process. Muscle development includes the formation of muscle fibers in the embryonic stage, the development of muscle fibers in the postnatal stage and the regeneration of muscle in the adult stage. This process is regulated at the transcriptional, posttranscriptional and levels [2].

The Duroc pig(D), which originated in the United States, is a typical lean meat-type pig. It has the advantages of fast growth, high meat yield, high feed conversion efficiency and excellent performance; however, it still has the disadvantages of poor meat quality [3]. The Duroc pig is the main terminal male breed in the pig industry worldwide [4,5]. The Luchuan pig(L) is an excellent local pig in China. It is a typical fat pig with the advantages of coarse feeding resistance, high temperature tolerance, high reproductive performance and good meat quality; however, its growth speed is slow, its fat content is high, and its meat yield is low [6,7]. Therefore, transcriptome sequencing of the longissimus dorsi muscle of these two breeds is of great significance in studying the growth traits and meat quality of pigs.

The growth and development of muscle is a complex process, including the formation of muscle fibers in the embryonic stage, the development of muscle fibers after birth and the regeneration of muscle in the adult stage. The embryonic stage mainly controls the formation of the individual muscle skeleton [8]. After birth, changes in different types of muscle fibers occur, such as increases in diameter and length; at the adult stage, damaged muscle fibers are repaired [9]. There are four types of muscle fibers in skeletal muscle [10]. The proportion of the muscle fiber type will directly affect meat quality [11]. Therefore, the study of its growth and development affects meat quality. Previous reports have shown that muscle fibers can be divided into red and white fibers [12]. These differences in fiber components determine their different metabolic types and physiological functions, as well as affect meat quality [13,14]. The content of intramuscular fat is positively correlated with the content of type I muscle fiber [15], the more type I muscle fibers in the muscle, the more fat content in the muscle, so the more tender the pork meat will be. For example, Yorkshire pigs have a higher percentage of type I muscle fibers than Hampshire pigs. Thus, Yorkshire pork is more tender [16]. The proportion of type I muscle fiber in the skeletal muscle of local Chinese breeds such as the Jinhua Pig and Beijing black pig is higher than that of foreign imported breeds, and the tenderness of these local breeds is also better than that of imported breeds [17].

Myosin light chain-2 (MYL2/MLC-2) is a small 167-amino acid protein of the myosin light chain family that is a regulatory light chain [18]. Basic light chains play an important role in the maintenance of heavy chain configuration, while regulatory light chains play a regulatory role in the activity of muscle fibers. Therefore, the proportion of four myosin light chains plays an important role in the type and growth of muscle fibers [19]. Upstream of the start codon of the skeletal muscle MYL2 promoter, multiple binding sites (MEF2, MyoD, and MyoG) that can promote the proliferation and differentiation of skeletal muscle cells are predicted. The MYL2 gene may have certain effects on the growth and differentiation of skeletal muscle cells [20,21]. The expression level of the MYL2 gene in the skeletal muscle of Duroc and white pigs was significantly higher than that in the local breed Piau, indicating that the gene is related to the growth of mammalian skeletal muscle [22]. In conclusion, the MYL2 gene may affect early skeletal muscle differentiation and participate in the growth and development of animal skeletal muscle, and the specific molecular regulatory mechanism needs further study.

In the present study, RNA-seq technology was employed to create expression profiles of the longissimus dorsi muscle tissue from young Luchuan pigs and Duroc pigs. Subsequently, differentially expressed genes (DEGs) were identified and subjected to Gene Ontology (GO) and pathway analyses to reveal the changes in gene expression and regulation that may be related to the differences in muscular development and meat quality traits between the breeds. This study mainly discusses the candidate genes that affect muscle growth, providing important regulatory information for the molecular mechanism of modern and local pork quality.

## 2. Materials and Methods

### 2.1. Ethics Statement and Collection of Tissue Samples

All the animal procedures used in this study were carried out according to the Guide for Care and Use of Laboratory Animals (8th edition, released by the National Research Council, Washington, DC, USA) and were approved by the Institutional Animal Care and Use Committee (IACUC) of Guangxi University. All the pigs in this study were sacrificed after anesthesia, and tissue samples (heart, liver, spleen, lung, kidney, longissimus dorsi and subcutaneous fat) were obtained, flash frozen in liquid nitrogen and stored at −80 °C. Each variety in different developmental periods (two months and eight months) contained four pigs (two males and two females), which were provided by The Animal Husbandry Research Institute of Guangxi Zhuang Autonomous Region. The longissimus dorsi of the 2-month-old Luchuan pig and Duroc pig were selected for RNA sequencing. The tissue expression profiles are used with different tissues (heart, liver, spleen, lung, kidney, longissimus dorsi and subcutaneous fat) from 2-month-old Luchuan pigs and Duroc pigs. The RNA used in the test was extracted by the kit method. Each trial was four biological replicates.

### 2.2. RNA Extraction, Library Construction and Illumina Sequencing

RNA degradation and contamination were monitored on 1% agarose gels. RNA purity was checked using the NanoPhotometer^®^ spectrophotometer (IMPLEN, Westlake Villag, CA, USA). The RNA concentration was measured using the Qubit^®^ RNA Assay Kit in a Qubit^®^ 2.0 Fluorometer (Life Technologies, Carlsbad, CA, USA). RNA integrity was assessed using the RNA Nano 6000 Assay Kit of the Bioanalyzer 2100 system (Agilent Technologies, Santa Clara, CA, USA). Three micrograms of RNA per sample were used as input material for the RNA sample preparations. Sequencing libraries were generated using the NEB Next^®^ Ultra™ RNA Library Prep Kit for Illumina^®^ (NEB, Ipswich, MA, USA) following the manufacturer’s recommendations, and index codes were added to attribute sequences to each sample. The library preparations were sequenced on an Illumina HiSeq platform, 125 bp/150 bp paired-end reads were generated, and the clean reads were aligned to the reference genome using TopHat v2.0.12. HTSeq v0.6.1 was used to count the read numbers mapped to each gene, and then the FPKM of each gene was calculated based on the length of the gene and read counts mapped to this gene.

### 2.3. Gene Ontology and Pathway Enrichment Analysis

Differential expression analysis of two conditions/groups (two biological replicates per condition) was performed using the DESeq R package (1.18.0). The *p*-values were adjusted using the Benjamini & Hochberg method. The corrected *p*-value of 0.005 and log2 (fold change) of 1 were set as the thresholds for significantly differential expression. Gene Ontology (GO) enrichment analysis of differentially expressed genes was implemented using the GOseq R package. GO terms with a corrected *p*-value < 0.05 were considered significantly enriched by differentially expressed genes. Kyoto Encyclopedia of Genes and Genomes (KEGG) pathway enrichment analysis of differentially expressed genes was carried out using the KEGG seq R package. KEGG pathways with corrected *p*-value < 0.01 were considered significantly enriched by differentially expressed genes.

### 2.4. Real–Time Reverse Transcription–Quantitative Polymerase Chain Reaction (RT-qPCR)

Total RNA was extracted from cells or tissues with TRIzol Reagent (Thermo Fisher Scientific, Waltham, MA, USA), followed by the synthesis of cDNA with reverse transcriptase and oligo-dT primers according to the manufacturer’s instructions (TaKaRa, Kusatsu, Japan). The 2^−ΔΔCt^ method was used to evaluate the quantitative variation, and the GAPDH gene was used as an internal control. Diluted cDNA was amplified using gene-specific primers (Table 1) and the TB Green real-time PCR master mix (TaKaRa, Japan). RT–PCR was used to verify the accuracy of the RNA-Seq data and detect the mRNA expression of the MYL2 gene in tissues and C2C12 cells, at least in triplicate with specific paired primers.

### 2.5. Vector Construction and Transient Transfection

The expression vector for the MYL2 gene was generated by the nest PCR-based cloning method to improve the specificity of the PCR product. Briefly, the complete coding sequence of MYL2 was inserted into the pEGFP-N1 vector, and the recombined vector was named pEGFP-N1-MYL2. First, the longer fragment containing the complete coding sequence of MYL2 was amplified from the longissimus dorsi muscle tissue with specially paired primers and Taq enzyme (Takara, Japan), and then the complete coding sequence of the target gene was amplified with special primers with restriction enzyme cutting sites and protecting bases. The specific primer pairs for gene cloning are listed in Table 1. Next, the pMD™18-T vector, E. coli DH5α competent cells, and HindIII and BamHI restriction enzymes (Takara, Japan) were used in vector construction. C2C12 cells were cultured in Dulbecco’s modified Eagle’s medium (DMEM) containing 10% fetal bovine serum (FBS) (Thermo Fisher Scientific, USA). The cells were transiently transfected with ~1 μg of the vector using Lipofectamine 3000 Reagent kit according to the manufacturer’s recommendations (Thermo Fisher Scientific, USA) when the density became 60%–70% in a 6-well cell culture plate. The fluorescence was observed in the cells after transfection for 48 h. Subsequently, the cells were harvested for RNA extraction, and RT-qPCR was carried out to detect the mRNA expression of relative genes.

### 2.6. Statistical Analysis

The SPSS statistical software package for Windows, Release 11.0.1 (15 November 2001, SPSS Inc., Chicago, IL, USA) and GraphPad Prism version 6.00 for Windows (La Jolla, CA, USA) were used for data analyses. In each experiment, the data were presented as means ± SD. Similar results were obtained in independent experiments. A paired sample *t*-test was used to analyze the differences between samples. The values are denoted as significant at *p* < 0.05.

## 3. Results

### 3.1. Summary of RNA-Seq Data

Eight RNA samples were prepared from the longissimus dorsi of Luchuan and Duroc pigs. These samples were sequenced using the Illumina HiSeq™ 2000 platform. After the unqualified low-quality reads, including adaptions or heights of unknown base N, were removed from the total raw reads, approximately 551 million total clean reads (Table 2), representing 90.33% of the total raw reads, were obtained. The average clean reads in Q20 and Q30 were 98.35% and 93.86%, respectively. When clean reads were mapped to the *Sus scrofa* genome, the total mapping ratios were above 90.56%, and the average unique mapping ratio was 66.58% (Table 2). Those indexes of the RNA-seq basic data confirmed that the data were high quality and could be used for further biological analysis.

### 3.2. Identification of Differentially Expressed Genes

Genes with similar expression patterns often have similar functions or are involved in the same metabolic processes (pathways). Thus, clustering of genes with similar expression patterns is an analytical strategy that can contribute to the identification of the function of unknown genes or can help to characterize the unknown functions of known genes. In order to identify clusters with functional enrichment, hierarchical clustering was performed based on gene expression patterns (Figure 1A). The gene expression profile showed small differences between the two different pig breeds. The gene expression levels were calculated using the fragments per kb per million reads (FPKM) values. Differentially expressed genes (DEGs) between sample groups were defined using the fold change values of the normalized (FPKM) expression values. DEGs were filtered with |log2 (Fold Change)| > 1 and corrected q-value (padj) < 0.001. Compared with the Duroc pig, 3682 DEGs comprised 2399 upregulated DEGs and 1283 downregulated DEGs were identified in the Luchuan pig (Figure 1B).

### 3.3. GO Enrichment and KEGG Pathway Analyses

In order to further determine the functions of the DEGs, functional categorization of all the DEGs was performed using GO annotation. The annotated results were classified into three parts: biological process, cellular component and molecular function (Figure 2A). The top five of each part were shown as follows: for part (1) biological process: (i) cellular process; (ii) metabolic process; (iii) biological regulation; (iv) regulation of biological process, and (v) response to stimulus, and for part (2) cellular component: (i) cell; (ii) cell part; (iii) organelle; (iv) membrane; (v) organelle part; (3) molecular function: (i) binding; (ii) catalytic activity; (iii) molecular function regulator; (iv) signal transducer activity, and (v) molecular transducer activity.

KEGG analysis of DEGs was also performed. As shown in Figure 2B, the pathways were classified into six groups by function: cellular processes, environmental information processing, genetic information processing, human diseases, metabolism and organismal systems. At the same time, the pathways were also sorted by the correlativity of enrichment (Figure 2C). We obtained 63 pathways (*p* < 0.05), among which the top 30 pathways are shown in Table 3. Combined the analysis results of GO enrichment and KEGG pathway and previous studies, we chose differentially expressed genes (Table 4) as candidate genes for further study that were associated with the biological processes of muscle tissue development, muscle cell differentiation, skeletal muscle tissue regeneration, fatty acid oxidation and lipid metabolism.

### 3.4. Verification of the Accuracy of the RNA-Seq Data Using RT-qPCR

Eight candidate genes involved in muscle and fat were randomly selected to validate the accuracy of the RNA-Seq data by RT-qPCR (Figure 3). The expression patterns of these eight genes were consistent with the RNA-Seq data. These results suggest that the RNA-Seq data are credible and can be used for subsequent experiments. This finding suggests that the identified DEGs play major roles in affecting the meat quality and growth rate of Duroc and Luchuan pigs.

### 3.5. Analysis of the MYL2 Gene Expression Pattern

Most of the functional genes have tissue-specific expression, so we detected the mRNA expression of the MYL2 gene in different tissues of pigs. The expression of the MYL2 gene in different tissues of 2-month-old Luchuan pigs (Figure 4A) and Duroc (Figure 4B) pigs was similar. Functional genes are in a dynamic condition in the same tissue during different developmental and growing periods. The MYL2 gene expression level in the longissimus dorsi of Luchuan and Duroc pigs was significantly higher at 2 months than at 8 months (*p* < 0.05) (Figure 4C). The expression level of the MYL2 gene in the longissimus dorsi of Luchuan and Duroc pigs at 2 and 8 months of age was detected, respectively, by RT-qPCR. The results showed that the MYL2 gene expression level in the Luchuan pig was significantly higher than that in the Duroc pig in both periods (*p* < 0.05) (Figure 4D). The cause may be related to the difference in growth speed and muscle fiber type of different pigs in different growth stages.

### 3.6. Expression Trend of the MYL2 Gene and Genes Related to Muscle Development in C2C12 Cells

The MYL2 gene may be related to muscle development. We screened some downstream genes (MYH1, MYH2, MYH4, MYH7, MYOD, MEF2 and MSTN) of MYL2 and detected their expression in C2C12 cells. Next, we tested whether MYL2 could change the gene expression in C2C12 cells by transfecting the overexpression vector MYL2. Fluorescence could be observed in C2C12 cells transfected with the pEGFP-N1-MYL2 recombined plasmid and empty vector pEGFP-N1 but not in the negative control group (Figure 5A). We found that the expression of MYH1, MYH2, MYH4, MYH7, MyoD, and MEF2 was increased significantly, and the expression of MSTN was decreased significantly when the expression of the MYL2 gene was increased in C2C12 cells(*p* < 0.05) (Figure 5B).

## 4. Discussion

Differences exist in the muscle growth and meat quality between Chinese and Western pig breeds, but the molecular mechanism remains unclear. RNA-seq is an effective method to identify new genes and their potential functional characteristics. Differences in the gene expression profiles can unveil mechanisms underlying the biological activities of various genes. Previous studies have shown that some changes in muscle fiber type characteristics and metabolic potential can explain changes in meat quality [23].

In this study, The RT-qPCR results showed that MYL2 gene expression in the longissimus dorsi of Luchuan and Duroc pigs was significantly higher at 2 months than at 8 months. The MYL2 gene expression in the longissimus dorsi of two-month-old, six-month-old, eight-month-old and ten-month-old black cattle decreased gradually [24], a finding that is similar to the results of this study. The MYL2 gene may affect the early differentiation of skeletal muscle and participate in the growth and development of skeletal muscle. The number of muscle fibers is stable before birth. Studies have shown that MYL2 is involved in the production of prenatal muscle fibers, and the expression of genes in the skeletal muscle of Duroc and Large White pigs was significantly higher than that of Piau [22,25]. Our results showed that the MYL2 gene expression in Luchuan pigs was significantly higher than that in Duroc pigs at 2 and 8 months of age. The cause may be related to the different times at which the maximum growth speed of different pigs appears. Studies have shown that the maximum growth rate of Duroc pigs is 130–150 days [2,26], while that of Luchuan pigs is 30–60 days [27]. The composition of skeletal muscle fiber types directly affects muscle quality. The content of intramuscular fat is positively related to the content of type I muscle fiber [15]. The proportion of type I muscle fiber in the skeletal muscle of the Jinhua Pig and Beijing black pig in China is higher than that of “Du × Chang × Da” and other introduced breeds, and the tenderness of these local breeds is also better than that of the introduced breeds [17]. Studies have shown that the MYL2 gene regulates the activity of fast muscle fiber and transforms the type of muscle fiber [28]. Additionally, the quality of the Luchuan pig is better than that of the Duroc pig, possibly explaining why MYL2 gene expression in the longissimus dorsi of 8-month-old Luchuan pigs is significantly higher than that of 8-month-old Duroc pigs. In conclusion, the MYL2 gene may be involved in the formation of skeletal muscle, early differentiation of skeletal muscle and transformation of muscle fiber types.

The MYL2 promoter is a skeletal muscle-specific promoter. Multiple binding sites (MEF2, MyoD and MyoG) that can promote the proliferation and differentiation of skeletal muscle cells are predicted upstream of the initiation codon [20,21]. These include myofibril, muscle system process, structural constituent of muscle, motor activity and Wnt signaling pathway. MEF2 transcription factors are major regulators of muscle differentiation and have been recently involved in activity-dependent muscle fiber type remodeling. A role of MEF2 genes in the regulation of the fiber type profile in vivo is supported by the finding that the proportion of type 1 fibers is decreased in mice with muscle-specific knockout of Mef2c or Mef2d, but not Mef2a and is increased by the overexpression of an activated Mef2c (MEF2c-VP16) [29]. MyoD can activate the transcription of muscle genes through multiple channels, thus promoting the differentiation of myoblasts. Compared with other myogenic regulators, MyoD mainly plays a role in the process of myogenesis, and its expression plays an important role in maintaining the differentiation of myocytes. The loss of MyoD can cause the proliferation and differentiation of myoblasts to fail [30,31]. Myostatin (MSTN) is a secreted growth factor that is mainly expressed in skeletal muscle and can inhibit muscle growth [32]. In recent years, many candidate genes related to muscle growth and meat quality traits have been found in domestic and foreign studies [33,34]. MSTN and MyoG are considered the main candidate genes related to muscle growth and development in animals [35,36]. We overexpressed the MYL2 gene in C2C12 cells and found that the expression levels of MYH1, MYH2, MYH4, MYH7, MyoD and MEF2 genes increased significantly with the significant increase in the MYL2 gene expression, and MSTN decreased significantly. Therefore, the MYL2 gene may be involved in the regulation of muscle formation and development, as well as the transformation of muscle fiber types, affecting the growth and development of livestock.

Through transcriptome sequencing results, we also obtained some signaling pathways related to muscle growth and development, such as the PI3K-Akt signaling pathway. The PI3K-Akt signaling pathway can regulate the proliferation and differentiation of myosatellite cells and also has a certain impact on skeletal muscle regeneration. Studies have shown that IGF1 and IFG2 bind to their receptors to activate the proliferation and differentiation of satellite cells through the PI3K-Akt signaling pathway [37,38]. Elia et al. activated the PI3K-Akt signaling pathway during the induction of C2C12 cell differentiation and found that the expression of MyoG and MCK genes increased, which in turn promoted the differentiation of C2C12 cells. After electroacupuncture treatment of rats with neuroskeletal atrophy, it was found that the expression of PI3K protein increased, and the expression of PI3K and AKT genes also increased, which ultimately promoted protein synthesis and muscle hypertrophy [39].

## 5. Conclusions

In this study, transcriptomic analyses of the longissimus dorsi muscle tissues of two-month-old Duroc pigs and Luchuan pigs were performed using RNA-seq. We obtained a large number of differentially expressed genes and selected the MYL2 gene for exploration. Test results showed that this gene might have a certain impact on muscle growth and development.

## Figures and Tables

**Figure 1 genes-14-00132-f001:**
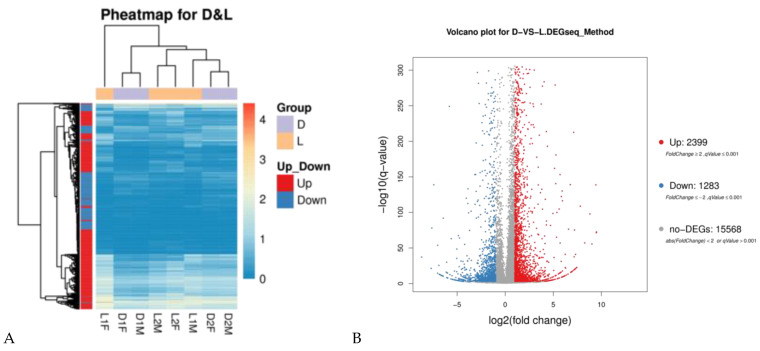
Transcriptome analysis of genes between the Duroc pig and Luchuan pig. (**A**) Heatmap analysis of gene expression based on the log10 ratio fold-change data. The horizontal axis represents samples, and the vertical axis represents differentially expressed genes that are upregulation and downregulation in red color and blue color, respectively. The color scale indicates gene expression based on log10 ratio fold-change data. (**B**) Volcano plot of significant differences in gene expression between the Duroc pig and Luchuan pig. Red spots represent the upregulation of genes, and blue dots are downregulated genes. Log2 FC indicates the fold change between the Duroc pig and the Luchuan pig. The longitudinal coordinates indicate the natural logarithm of the magnitude of the control groups. The longitudinal coordinates indicate the magnitude of differences at the transcriptional level.

**Figure 2 genes-14-00132-f002:**
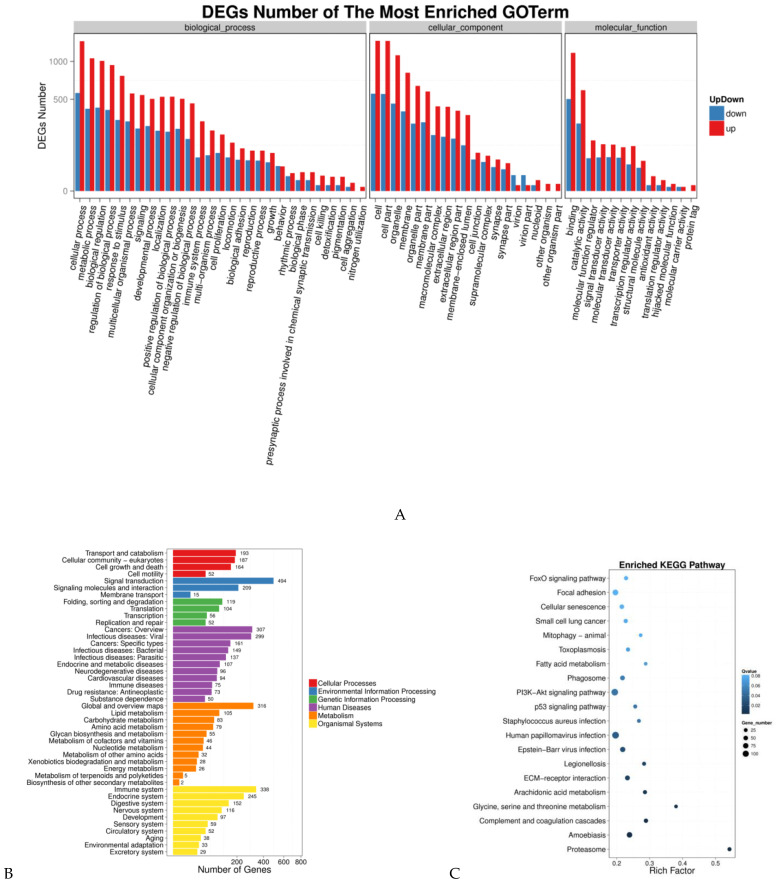
(**A**): Column diagrams for Gene Ontology (GO) analysis of DEGs. The *x*-axis represents the functions of GO analysis. The *y*-axis represents the number of DEGs. Red represents upregulated DEGs. Blue represents downregulated DEGs. (**B**): Column diagrams for Kyoto Encyclopedia of Genes and Genomes (KEGG) analysis of DEGs. The *x*-axis represents the number of DEGs. The *y*-axis represents the functions of pathways. Each color represents the appropriate biological process. (**C**): Diagrams for the enrichment degree of pathways. The *x*-axis represents the value of rich factors (the ratio of annotated DEGs to all genes of the enriched pathway). The *y*-axis represents the names of pathways. The color depth of each point represents the q value. The size of each point represents the number of DEGs.

**Figure 3 genes-14-00132-f003:**
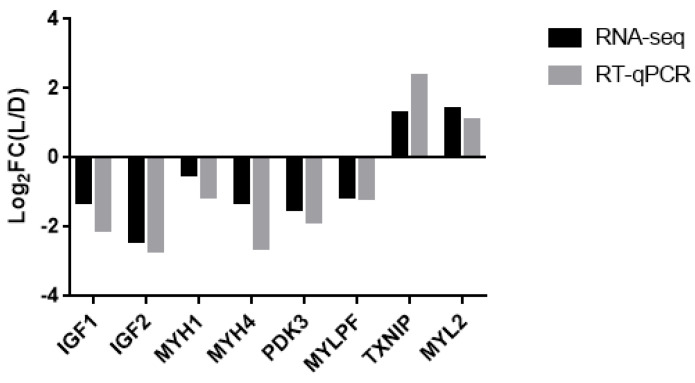
Log2FC (LD/DL) indicates the fold change between the Duroc pig and the Luchuan pig. The mRNA expression of different genes in the transcriptome is shown by the value of log2 ratio (LD/DL) to conveniently compare with the results of RT-qPCR. The Duroc pig represents the control group.

**Figure 4 genes-14-00132-f004:**
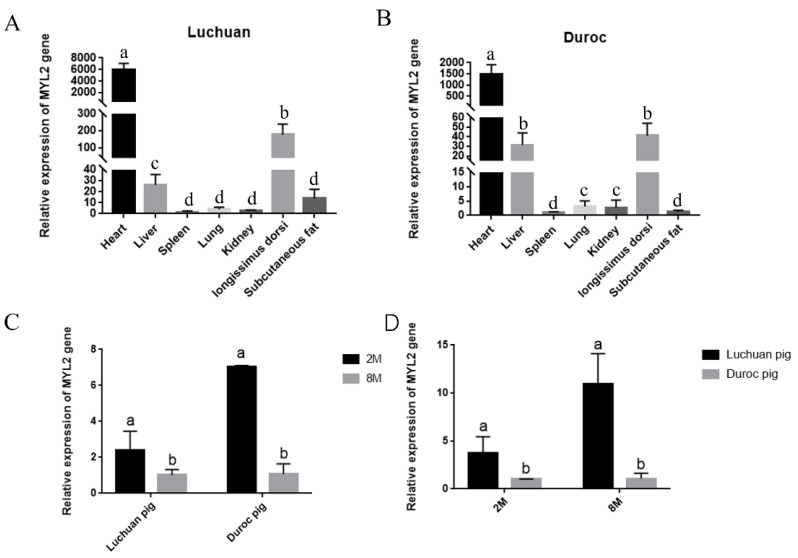
Expression pattern of the MYL2 gene. (**A**,**B**): MYL2 mRNA expression pattern in different tissues of two-month-old Duroc and Luchuan pigs. (**C**,**D**): MYL2 mRNA expression pattern in different pig breeds and developmental periods. Different letters indicate a significant difference (*p* < 0.05). All the results are expressed as means ± SD.

**Figure 5 genes-14-00132-f005:**
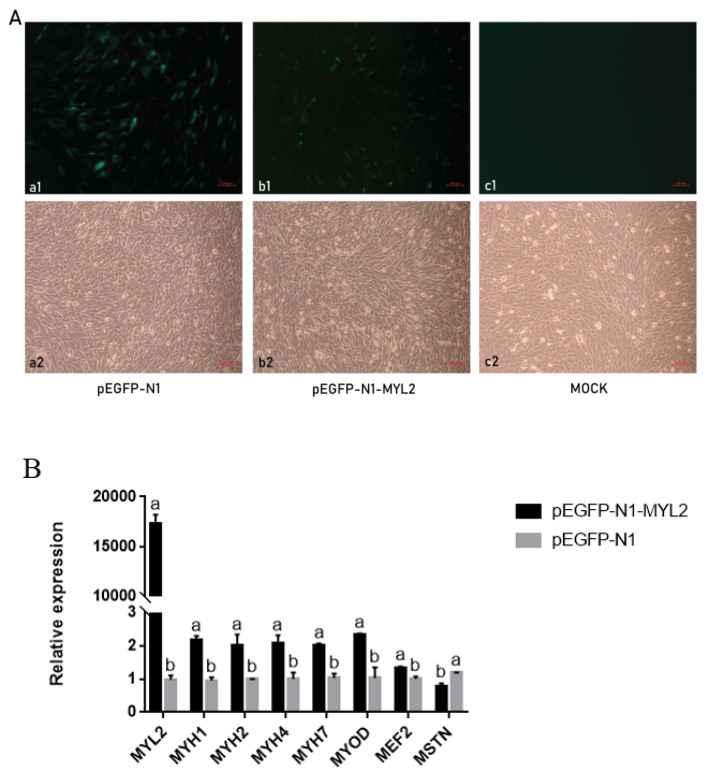
(**A**) The plasmid was transfected into C2C12 cells. (**a1**–**c1**), cells were observed under a fluorescence microscope at 10×; (**a2**–**c2**), cells observed under an optical microscope at 10×. (**a1**,**a2**), cells transfected with pEGFP-N1; (**b1**,**b2**), cells transfected with pEGFP-N1-MYL2; (**c1**,**c2**), blank control group. (**B**) The mRNA expression change of MYL2, MYH1, MYH2, MYH4, MYH7, MYOD, MEF2 and MSTN after transfecting an overexpression vector of MYL2 into C2C12 cells. Different letters indicate a significant difference (*p* < 0.05). All of the results are expressed as means ± SD.

**Table 1 genes-14-00132-t001:** Primer sequences for reverse transcription–quantitative PCR (RT-qPCR) and PCR.

Gene	Gene ID in NCBI	Primer Type	Primer Sequence (5′→3′)	Product Size (bp)
MYL2-sus	NM_213791.2	RT-qPCR	Forward: ACAGGGATGGCTTCATAGACAA	156
			Reward: TAAGTTTCTCCCCAAACATCGT	
IGF1-sus	NM_214256.1	RT-qPCR	Forward: GGACGCTCTTCAGTTCGTGT	117
			Reward: CGGAAGCAGCACTCATCCAC	
IGF2-sus	NM_213883.2	RT-qPCR	Forward: ACACCCTCCAGTTTGTCT	236
			Reward: GTCATAGCGGAAGAACTTGCC	
MYH1-sus	NM_001104951.2	RT-qPCR	Forward: GTGCCGTACTCTAGAAGATCAAC	158
			Reward: GCTGAGAAACTAACGTGTCCT	
MYH4-sus	NM_001123141.1	RT-qPCR	Forward: TACTTAGGAAGAAGCACGCAGAC	242
			Reward: ATTGATTAGGCGCTGATGCTC	
PDK3-sus	XM_021080538.1	RT-qPCR	Forward: CACTTCTCTTTGGCGGTGACA	120
			Reward: CACAAAGCATCTTGGCGGTTT	
MYLPF-sus	NM_001006592.1	RT-qPCR	Forward: CAGAAGGAAGCTCCAACGTCT Reward: GAGCTTCTCCCCAAACATGGTC	239
TXNIP-sus	NM_001044614.2	RT-qPCR	Forward: CTAGTGGATGTCAATACCCCT Reward: TTGCCTCTGACCGATGACAAC	278
GAPDH-sus	NM_001206359.1	RT-qPCR	Forward: AACATCATCCCTGCTTCTACCG	158
			Reward: CAGGTCAGATCCACAACCGACA	
MYH1-mus	NM_030679.2	RT-qPCR	Forward: GCAGCTCCAAGTTCAGTCT	289
			Reward: TGCCATCTCCTCTGTCAGGT	
MYH2-mus	NM_001039545.2	RT-qPCR	Forward: AAGCAGAGGCAAGTAGTGGT	124
			Reward: GTGCTCCTGAGATTGGTCAT	
MYH4-mus	NM_010855.3	RT-qPCR	Forward: GTTCTTCTTTCCAGACCGT	130
			Reward: ATGGCACCAGGAGTCTTAG	
MYH7-mus	NM_080728.3	RT-qPCR	Forward: GACAAAGGCAAAGGCAAGG	129
			Reward: GATGATGCAGCGTACAAAG	
MYOD-mus	NM_010866.2	RT-qPCR	Forward: TGGCATGATGGATTACAGC Reward: TTCCCTGTTCTGTGTCGCTT	260
MEF2c-mus	NM_001347567.1	RT-qPCR	Forward: CTCTGTCTGGCTTCAACACT Reward: CTGACTTGATGCTGAGGCTTT	183
MSTN-mus	NM_010834.3	RT-qPCR	Forward: ATGACGATTATCACGCTACCAC Reward: ACCTTGTACCGTCTTTCAT	234
MYL2-clone	NM_213791.2	PCR	Reward: CCATGTCACCTAAGAAAGCCAAG	549
			Reward: CCACTCACCCAAAGGCAAG	
MYL2-vc ^a^	NM_213791.2	PCR	Forward: CCCAAGCTTATGTCACCTAAGAAAGCCAAGAAG	503
			Reward: CGCGGATCCCCTTAGTCCTTCTCTTCTCCGTGG	

Note: ^a^ This primer pair is used in the amplification of the complete coding sequence of the MYL2 gene for vector construction. The letters with a single underline represent restriction enzyme cutting sites, and the letters with double underlines represent protecting bases. The primers marked ‘Sus,’ and ‘Mus’ were used for mRNA expression in pig tissues and C2C12 cells by real-time quantitative RT–PCR (RT-qPCR).

**Table 2 genes-14-00132-t002:** Summary of the sequencing data quality and statistics of the transcriptome.

Sample	D1F	D1M	D2F	D2M	L1F	L1M	L2F	L2M
Total Raw Reads (M)	75.32	73.68	77.02	77.01	77.75	80.23	75.32	73.69
Total Clean Reads (M)	72.65	71.8	66.57	66.44	64.23	65.15	72.85	71.35
Total Clean Bases (Gb)	7.27	7.18	6.66	6.64	6.42	6.52	7.29	7.14
Clean Reads Q20 (%)	98.75	98.87	97.97	98.07	98.04	97.83	98.66	98.64
Clean Reads Q30 (%)	96.05	96.34	91.89	92.11	91.77	91.13	95.83	95.78
Clean Reads Ratio (%)	96.45	97.44	86.44	86.27	82.61	81.12	96.72	96.84
Total Mapping Ratio	91.79%	92.36%	95.39%	95.22%	93.05%	93.56%	91.10%	90.56%
Uniquely Mapping Ratio	68.94%	70.28%	67.05%	68.74%	63.74%	58.74%	67.56%	67.57%

Note: D indicates Duroc pig; L indicates Luchuan pig; F indicates female; M indicates male.

**Table 3 genes-14-00132-t003:** Significantly enriched KEGG pathways in RNA-Seq analysis.

	Pathway	DEGs Genes with Pathway Annotation (2884)	*p* Value	Pathway ID
1	Proteasome	32	2.74 × 10^−12^	ko03050
2	Amoebiasis	80	1.09 × 10^−5^	ko05146
3	Complement and coagulation cascades	41	1.68 × 10^−5^	ko04610
4	Glycine, serine and threonine metabolism	19	5.83 × 10^−5^	ko00260
5	Arachidonic acid metabolism	34	0.000107	ko00590
6	ECM-receptor interaction	60	0.000279	ko04512
7	Legionellosis	30	0.000316	ko05134
8	Epstein-Barr virus infection	72	0.000551	ko05169
9	Human papillomavirus infection	121	0.000915	ko05165
10	Staphylococcus aureus infection	27	0.001579	ko05150
11	p53 signaling pathway	31	0.00158	ko04115
12	PI3K-Akt signaling pathway	117	0.001723	ko04151
13	Phagosome	57	0.002097	ko04145
14	Fatty acid metabolism	19	0.003029	ko01212
15	Toxoplasmosis	37	0.003322	ko05145
16	Mitophagy-animal	21	0.003851	ko04137
17	Small cell lung cancer	40	0.004097	ko05222
18	Cellular senescence	50	0.004594	ko04218
19	Focal adhesion	87	0.004764	ko04510
20	FoxO signaling pathway	37	0.00515	ko04068
21	MAPK signaling pathway	79	0.005275	ko04010
22	Hematopoietic cell lineage	35	0.00558	ko04640
23	DNA replication	19	0.006139	ko03030
24	PPAR signaling pathway	28	0.00717	ko03320
25	Transcriptional misregulation in cancer	64	0.007257	ko05202
26	Fc epsilon RI signaling pathway	26	0.008615	ko04664
27	Platinum drug resistance	25	0.008867	ko01524
28	Pathways in cancer	150	0.009306	ko05200
29	Amyotrophic lateral sclerosis (ALS)	23	0.009324	ko05014
30	Gastric cancer	56	0.009609	ko05226

**Table 4 genes-14-00132-t004:** Candidate genes related to growth and development in D-vs-L.

GeneID	log2Ratio (L/D)	q-Value	Up-Down-Regulation (L/D)	*p*-Value	Symbol
100144306	−2.585066	0	Down	0	*MYH4*
100125538	−1.110562	0	Down	0	*MYH1*
474162	−1.163615	0	Down	0	*MYLPF*
396725	1.20822	0	Up	0	*DES*
396718	−1.080459	0	Down	0	*RYR1*
100286778	7.811893	0	Up	0	*PDK4*
397667	1.096899	0	Up	0	*FHL1*
733657	5.046722	0	Up	0	*FBXO32*
396916	−2.672299	0	Down	0	*IGF2*
733688	2.369902	0	Up	0	*TXNIP*
100152001	−1.666755	0	Down	0	*COL3A1*
414388	1.804317	0	Up	0	*TPM3*
100517321	3.11986	0	Up	0	*TSC22D3*
396690	1.081174	0	Up	0	*MYL2*
100525195	2.038505	0	Up	0	*LMOD2*
100271745	1.007136	0	Up	0	*TCAP*
396947	1.090353	0	Up	0	*TNNI1*
397583	4.714497	0	Up	0	*LIPE*
397077	2.081716	0	Up	0	*FOXO1*
780431	1.834607	0	Up	0	*NDRG2*
396959	1.35284	0	Up	0	*ANKRD1*
100156435	1.058298	0	Up	0	*TNNC1*
100048933	1.669136	0	Up	0	*ZFAND5*
397116	3.909268	0	Up	0	*UCP3*
100294702	−1.503496	0	Down	0	*SMYD1*
100157318	3.569646	0	Up	0	*APOD*
100624868	1.763672	0	Up	0	*BNIP3*
100519789	1.258786	0	Up	0	*CRYAB*
100152767	−1.115181	0	Down	0	*KLHL31*
100627008	−1.382584	0	Down	0	*DMPK*
100520636	1.633977	0	Up	0	*ACADVL*
733702	1.161209	0	Up	0	*CD36*
100134962	1.63106	0	Up	0	*PHYH*
100337687	1.69399	0	Up	0	*CSRP3*
100240723	1.662207	0	Up	0	*SGCA*
100620966	4.171741	0	Up	0	*SESN2*
100157793	−1.276461	0	Down	0	*PRKAB2*
100152267	−2.014984	0	Down	0	*ACTC1*
100152836	1.60654	0	Up	0	*GPAT4*
397012	1.111651	0	Up	0	*HADHA*

## Data Availability

The high-throughput sequencing raw data from this study have been submitted to the OSF (https://osf.io/, accessed on 4 November 2022, Duroc and Luchuan pigs). Named Duroc and Luchuan pigs.

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
