# Peer review of "Identification of Differentially Expressed Genes in the Longissimus Dorsi Muscle of Luchuan and Duroc Pigs by Transcriptome Sequencing"

_genes, 2023, doi:10.3390/genes14010132_

Round 1

Reviewer 1 Report

1. Lines 15 and 43 assert that the meat quality of the Duroc pig is poor but there are no citations/references to support the assertion.

2. In Line 219 there are eight candidate genes yet in Line 220 we have the expression of these SEVEN genes. I think there is an error regarding one of the numbers.

Author Response

Dear reviewers,

Thank you for your letter and for the reviewers’ comments concerning our manuscript entitled “Identification of differentially expressed genes in the longis-simus dorsi muscle of Luchuan and Duroc pigs by transcrip-tome sequencing” (genes-2046411). Those comments were all valuable and very helpful for revising and improving our paper, as well as for providing important guiding significance to our study. We have carefully revised our manuscript according to your and reviewers’ comments. Revised portion are marked in red in the paper. Here, we listed the point-by-point responses to your detailed comments and suggestions (With red). As follows:

  1. Lines 15 and 43 assert that the meat quality of the Duroc pig is poor but there are no citations/references to support the assertion.

Response: References have been added to the revised manuscript.

  1. In Line 219 there are eight candidate genes yet in Line 220 we have the expression of these SEVEN genes. I think there is an error regarding one of the numbers.

Response: It should be 8, which has been revised in the revised version.

Reviewer 2 Report

Dear authors,

The manuscript entitled “Identification of differentially expressed genes in the longissimus dorsi muscle of Luchuan and Duroc pigs by transcriptome sequencing” aims to discuss the candidate genes that affect muscle growth, providing important regulatory information for the molecular mechanism of modern and local pork quality. The manuscript has a good abstract and introduction, and brings interesting results, however, it needs many corrections in Material and Methods, Discussion and Conclusion. The design of the work is confusing, there is a lack of information about animal management, the genes evaluated by RT-qPCR, the reason for analyzing the expression of these genes in C2C12 cells. The discussion superficially approaches the main focus of the work, which is the information regarding the analysis of gene ontology and KEGG using differentially expressed genes to understand phenotypic differences related to the growth of pigs of the Duroc and Luchuan breeds. The conclusion does not focus on the results obtained based on the objective of the study. It needs major changes and corrections. Below are suggestions to make the manuscript more informative to the reader.

 General suggestions

Title: Correct “longissimus dorsi” to “Longissimus dorsi”.

Abstract

Line 14: Removing "and" after United States;

Line 20: Write the name of the MYL2 gene (Myosin light chain-2).

Keywords: Arrange in alphabetical order.

 Introduction

Lines 39-40: Please remove the word “pathway” from the sentence.

Line 55: separate “at” and “the”.

Line 61-64: Please rewrite this sentence to make it more understandable to the reader, it's too long.

Line 51-68: Paragraph too long.

Line 69: Remove the first comma from that sentence.

Line 85: Add “differentially expressed genes” before “DEGs”.

 Materials and Methods

Information about experimental design, animal handling, number of animals in each analysis, which tissue was used in each analysis should be added.

Line 97: Change “longissimus dorsi” to “Longissimus dorsi”.

Line 92-101: Did you work with 2 and 8 month old animals in each analysis? Please make clear the experimental design used.

Line 102: Add a short information about RNA extraction. Please write about which tissue was used for the RNA extraction and RNAseq analyses, as you mentioned collecting several different tissues.

Information on the number of animals per genetic group used for RNAseq is required. Explain more about “treatments” in the Materials and Methods.

Alter “longissimus dorsi” to “Longissimus dorsi” in all text.

How many genes were evaluated in Longissimus dorsi and in C2C12 cells by RT-qPCR? Make it clear why these genes were chosen.

 Results

Line 170: Did you work with a N of four Duroc animals and a N of four Luchuan animals? Were these animals 2 and 8 months old? As I said earlier, make your experimental design more understandable for the reader in the Material and Methods.

Line 175: Alter “Sus scrofa” to “Sus scrofa”.

Line 211-215: What are these additional studies in relation to these candidate genes?

Did you perform the RT-qPCR of the MYH7, MYOD and MEF2C genes? Why are they in the primers table but not in the results figure?

Why was the IGF1 gene chosen for RT-qPCR analysis since it is not found in the table of candidate genes? Make it clear in the text how these candidate genes were determined.

Line 219-221: “Eight candidate genes involved in muscle and fat were randomly selected to validate the accuracy of the RNA-Seq data by RT–qPCR (Fig.3). The expression patterns of these 7 genes were consistent with the RNA-Seq data.” Was it eight or seven genes?

Line 227-229: “The expression of the MYL2 gene in different tissues of 2-month-old Luchuan pigs (Fig. 4A) and Duroc (Fig. 228 4B) pigs was similar.” Add the p-value. Was the statistical analysis performed between tissues considering one breed at a time? A comparison between tissues considering the Duroc breed and a comparison between tissues considering the Luchuan breed?

Line 230-232: “The MYL2 gene expression level in the longissimus dorsi of Luchuan and Duroc pigs was significantly higher at 2 months than at 8 months (p< 0.05) (Fig.4C).” Here comparisons were made between ages within each race?

Line 234-235: “The results showed that the MYL2 gene expression level in the Luchuan pig was significantly higher than that in the Duroc pig in both periods (p< 0.05)” Comparison between ages within each age? The results are mixed as they look like different designs.

Line 239-241: Some genes were analyzed by RT-qPCR only in C2C12 cells. Please, separate in the primers table the primers that were used for analysis of the collected tissues and the primers that were used for analysis of the cells. I suggest putting this information in Material and Methods.

 Discussion

The discussion is superficial and focused only on the MYL2 gene. You did not discuss the observed results of the analyses of gene ontology and KEGG of DEGs to discuss the factors that determine growth in the Longissimus dorsi muscle of Luchuan and Duroc pigs. A much deeper discussion of the results (candidate genes, biological processes/ pathways…) should be carried out to meet the objective of the study.

Line 256: Alter “difference” to “differences”.

Line 255-284: Rewrite this paragraph please, it is too long.

Line 290:  Correct  “elevance” to “relevance”.

Line 285-320: Rewrite this paragraph please, it is too long.

Conclusion

The conclusion does not focus on the results obtained based on the objective of the study (Differences between breeds considering transcriptome sequencing).

Author Response

Dear reviewers,

Thank you for your letter and for the reviewers’ comments concerning our manuscript entitled “Identification of differentially expressed genes in the longis-simus dorsi muscle of Luchuan and Duroc pigs by transcrip-tome sequencing” (genes-2046411). Those comments were all valuable and very helpful for revising and improving our paper, as well as for providing important guiding significance to our study. We have carefully revised our manuscript according to your and reviewers’ comments. Revised portion are marked in red in the paper. Here, we listed the point-by-point responses to your detailed comments and suggestions (With red). As follows:

  1. Title: Correct “longissimus dorsi” to “Longissimus dorsi”.

Response: It has been revised in the revised draft.

  1. Abstract

Line 14: Removing "and" after United States;

Response: It has been revised in the revised draft.

Line 20: Write the name of the MYL2 gene (Myosin light chain-2).

Response: Added in revised draft.

3.Keywords: Arrange in alphabetical order.

Response: It has been revised in the revised draft.

4.Introduction

Lines 39-40: Please remove the word “pathway” from the sentence.

Response: It has been deleted in the revised version.

Line 55: separate “at” and “the”.

Response: It has been revised in the revised draft.

Line 61-64: Please rewrite this sentence to make it more understandable to the reader, it's too long.

Response: Corresponding modifications and deletions have been made.

Line 51-68: Paragraph too long.

Response: The literature cited is a comparison of Chinese endemic breeds and foreign pig breeds, and it feels that not deleting them can better help readers understand.

Line 69: Remove the first comma from that sentence.

Response: It has been deleted in the revised version

Line 85: Add “differentially expressed genes” before “DEGs”.

Response: It has been added in the revised version.

5.Materials and Methods

Information about experimental design, animal handling, number of animals in each analysis, which tissue was used in each analysis should be added.

Response: This has been explained in the revised draft

Line 97: Change “longissimus dorsi” to “Longissimus dorsi”.

Response: It has been revised in the revised draft.

Line 92-101: Did you work with 2 and 8 month old animals in each analysis? Please make clear the experimental design used.

Response: The usefulness of the organization has been added in the revised version.

Line 102: Add a short information about RNA extraction. Please write about which tissue was used for the RNA extraction and RNAseq analyses, as you mentioned collecting several different tissues.

Response: This has been added to 2.1 in the revised version.

Information on the number of animals per genetic group used for RNAseq is required. Explain more about “treatments” in the Materials and Methods.

Response: This has been added to 2.1 in the revised version.

Alter “longissimus dorsi” to “Longissimus dorsi” in all text.

Response: There are 17 places in the full text, all of which have been revised.

How many genes were evaluated in Longissimus dorsi and in C2C12 cells by RT-qPCR? Make it clear why these genes were chosen.

Response: We obtained 3682 differentially expressed genes, and then selected the MYL2 gene after screening and reviewing a large number of literature, and finally performed some functional verification on C2C12 cells.

6.Results

Line 170: Did you work with a N of four Duroc animals and a N of four Luchuan animals? Were these animals 2 and 8 months old? As I said earlier, make your experimental design more understandable for the reader in the Material and Methods.

Response: This has been added to 2.1 in the revised version.

Line 175: Alter “Sus scrofa” to “Sus scrofa”.

Response: It has been revised in the revised draft.

Line 211-215: What are these additional studies in relation to these candidate genes?

Response: These genes are obtained by consulting some literature.

Did you perform the RT-qPCR of the MYH7, MYOD and MEF2C genes? Why are they in the primers table but not in the results figure?

Response: These three genes are some of the hallmark genes for muscle growth and development, which were detected after cell testing, and are available in Figure 5B.

Why was the IGF1 gene chosen for RT-qPCR analysis since it is not found in the table of candidate genes? Make it clear in the text how these candidate genes were determined.

Response: Figure 3 was randomly selected among 3682 differential genes, and the genes in Table 4 were selected and displayed among 3682 genes after I consulted a certain amount of literature.

Line 219-221: “Eight candidate genes involved in muscle and fat were randomly selected to validate the accuracy of the RNA-Seq data by RT–qPCR (Fig.3). The expression patterns of these 7 genes were consistent with the RNA-Seq data.” Was it eight or seven genes?

Response: It is 8 and has been revised in the revised version.

Line 227-229: “The expression of the MYL2 gene in different tissues of 2-month-old Luchuan pigs (Fig. 4A) and Duroc (Fig. 228 4B) pigs was similar.” Add the p-value. Was the statistical analysis performed between tissues considering one breed at a time? A comparison between tissues considering the Duroc breed and a comparison between tissues considering the Luchuan breed?

Response: The p-value has been added to the plot.

Because the focus of the article is on muscle, only different varieties and months of age are compared for muscle tissue

Line 230-232: “The MYL2 gene expression level in the longissimus dorsi of Luchuan and Duroc pigs was significantly higher at 2 months than at 8 months (p< 0.05) (Fig.4C).” Here comparisons were made between ages within each race?

Response: Figure 4D compares the different months of age of each breed of pig.

Line 234-235: “The results showed that the MYL2 gene expression level in the Luchuan pig was significantly higher than that in the Duroc pig in both periods (p< 0.05)” Comparison between ages within each age? The results are mixed as they look like different designs.

Response:

Line 239-241: Some genes were analyzed by RT-qPCR only in C2C12 cells. Please, separate in the primers table the primers that were used for analysis of the collected tissues and the primers that were used for analysis of the cells. I suggest putting this information in Material and Methods.

Response: The primer suffix clearance "sus" and "mus" are distinguished.

7.Discussion

The discussion is superficial and focused only on the MYL2 gene. You did not discuss the observed results of the analyses of gene ontology and KEGG of DEGs to discuss the factors that determine growth in the Longissimus dorsi muscle of Luchuan and Duroc pigs. A much deeper discussion of the results (candidate genes, biological processes/ pathways…) should be carried out to meet the objective of the study.

Response: Thank you very much for your guidance, since there was no discussion without doing the study of the signaling pathway, we will further deepen the research in this area in the future.

Line 256: Alter “difference” to “differences”.

Response: It has been revised in the revised draft.

Line 255-284: Rewrite this paragraph please, it is too long.

Response: Thanks for the guidance, we have made changes, please look at lines 256 to 280.

Line 290:  Correct  “elevance” to “relevance”.

Response: It has been revised in the revised draft.

Line 285-320: Rewrite this paragraph please, it is too long.

Response: Thanks for the guidance, we have made changes, please look at lines 281 to 304.

8.Conclusion

The conclusion does not focus on the results obtained based on the objective of the study (Differences between breeds considering transcriptome sequencing).

Response: Some changes have been made in the revised draft.
